# Investigation of the Indications for Endoscopic Papillectomy and Transduodenal Ampullectomy for Ampullary Tumors

**DOI:** 10.3390/jcm10194463

**Published:** 2021-09-28

**Authors:** Masanari Sekine, Fumiaki Watanabe, Takehiro Ishii, Takaya Miura, Yudai Koito, Hitomi Kashima, Keita Matsumoto, Hiroshi Noda, Toshiki Rikiyama, Hirosato Mashima

**Affiliations:** 1Saitama Medical Center, Department of Gastroenterology, Jichi Medical University, Saitama 330-8503, Japan; take3546@jichi.ac.jp (T.I.); tmiura0630@gmail.com (T.M.); koito0406@yahoo.co.jp (Y.K.); hitomi.19881206@gmail.com (H.K.); wrcr1007@gmail.com (K.M.); hmashima@jichi.ac.jp (H.M.); 2Saitama Medical Center, Department of Surgery, Jichi Medical University, Saitama 330-8503, Japan; fwatanabe210@yahoo.co.jp (F.W.); noda164@omiya.jichi.ac.jp (H.N.); trikiyama@jichi.ac.jp (T.R.)

**Keywords:** endoscopic papillectomy, transduodenal ampullectomy, ampullary tumors, adenoma, adenocarcinoma

## Abstract

Objective: The standard treatment for ampullary tumors is pancreaticoduodenectomy. However, minimally invasive procedures such as endoscopic papillectomy (EP) and transduodenal ampullectomy (TDA) have recently gained popularity. Therefore, we aimed to evaluate the effectiveness of these minimally invasive procedures for ampullary tumors. Methods: We conducted a retrospective study of 42 patients who underwent either EP or TDA for ampullary tumors between June 2011 and November 2020. Results: We found that in patients with significantly larger tumors, TDA was often selected. Patients who underwent EP had significantly shorter hospital stays. No significant differences were observed regarding procedural accidents, tumor size, and recurrence. Conclusion: No differences were observed regarding the treatment outcomes of EP and TDA except hospital stay. EP is less invasive and can be the initial choice of procedure. TDA is performed when EP is not technically feasible. No significant relationship was noted between tumor size and recurrence, and careful observation of the patient’s postoperative course is required.

## 1. Introduction

Pancreaticoduodenectomy (PD) has been commonly performed to manage ampullary tumors regardless of malignancy status. However, PD is associated with the high degree of invasiveness. In 1983, the first report on endoscopic papillectomy (EP) by Suzuki et al. [1] was published. Subsequently, it was widely used despite a high-risk treatment, but it has not become the standard treatment yet. Contrastingly, the first report of transduodenal ampullectomy (TDA) was published in 1899 by Halsted. [2] However, consensus regarding its indications remain controversial due to its high recurrence rate. 

The European Society of Gastrointestinal Endoscopy (ESGE) guidelines for ampullary tumors have recently been reported, which stipulates that the indication for EP is high-grade dysplasia with a size between 20 and 30 mm and bile or pancreatic duct progression measuring ≤20 mm [3]. Conversely, the indication for TDA includes Tis cancer, adenoma demonstrating bile or pancreatic duct progression measuring >20 mm, and adenoma wherein EP would present with technical difficulties due to diverticulum or a large size measuring ≥40 mm. Systematic review with meta-analysis reported an increased rate of complete resection in surgical interventions (PD, TDA), accompanied with a high risk of complications (PD), and no significance in recurrence between EP and TDA [4].

EP was reported to be associated with increased risk of remnants, but its outcome is improving with the progress of the equipment. TDA is a more radical treatment but is associated with a high degree of invasiveness. In a society where the population is aging rapidly like Japan, it is important to evaluate whether less invasive EP or more radical TDA was more effective for ampullary adenomatous lesions. In this study, we compared and evaluated the effectiveness of EP and TDA for the treatment of ampullary tumors. 

## 2. Methods

We conducted a retrospective study of 42 patients who underwent EP or TDA as the initial treatment for ampullary tumors at Saitama Medical Center, Jichi Medical University, between June 2011 and November 2020. 

The information of patients was retrieved from medical records. Definition of mortality is 30 days mortality.

### 2.1. Preoperative Tests

All subjects were observed, and biopsies were performed using a rear oblique-view scope (JF260V, TJF260, TJF290, Olympus Corp, Tokyo, Japan). Endoscopic ultrasonography (EUS) and/or endoscopic retrograde cholangiopancreatography was used to observe and assess the T factor and superficial bile or pancreatic duct progression. Multidetector computed tomography (MD-CT) scan was used to assess N and M factors. During the study period, no clear guidelines regarding ampullary tumors have been detailed; therefore, the attending physicians discussed and determined the choice of EP or TDA. As a general rule, the target was adenoma lesions; however, a small number of patients with adenocarcinoma were included. PD was selected when the patient was positive for bile or pancreatic duct progression, T2 or deeper invasion, or positive N-factor.

### 2.2. Treatment Details

#### 2.2.1. Endoscopic Papillectomy

All the procedures were performed under intravenous anesthesia. All patients underwent evaluation using rear oblique view scopes (JF260V, TJF260, TJFQ290V, Olympus Corp.). After confirming the presence of the ampullary tumor, a margin was established around the tumor from the oral protrusion to the frenulum, and snaring was performed. Resection was performed using a high-frequency device (ICC200 Erbe Elektromedizin, Tubingen, Germany. ENDO CUT^®^ Effect3 cut 120 W coag 30 W, or ESG-100 Olympus Corp., Tokyo, Japan. Pulsecut-slow LEVEL30). The scope was removed temporarily. After collecting the specimen, the scope was reinserted, and the frenulum was sutured with clips. After bile and pancreatic duct cannulation, guidewire indwelling plastic stents (bile duct, 7 Fr., 5 or 7 cm; pancreatic duct, 5 Fr., 4, 7, or 9 cm) were installed in the bile and pancreatic ducts. Without conducting a second look, the rear oblique view scope was re-inserted 5–7 days after to evaluate the resection site; additionally, the stents were removed and a biopsy of the margin of the resected ulcer was performed. 

#### 2.2.2. Transduodenal Ampullectomy 

TDA was performed in all patients under general anesthesia. A Kocher maneuver was performed with duodenal mobilization and exposure of the posterior wall of duodenum. After palpation of the duodenum for the identification of the ampullary lesion, a 2–4 cm longitudinal duodenotomy was performed and the ampullary lesion was visualized (Figure 1A). Stay sutures were placed around the circumference of the tumor and physiological saline (5 cc) was injected into the submucosa to lift the lesion. The duodenal mucosa was incised at least 5 mm from tumor and ampulla tumor was resected with careful identification of the sphincter of Oddi (Figure 1B). To repair the cavity of the lost mucosa, the mucosa and the sphincter of Oddi were radially sutured to prevent obstruction of the Wirsung duct and the common bile duct (CBD). The duodenum wall was sutured in the direction of the short axis using the Gambee suture pattern. 

#### 2.2.3. Observation of Postoperative Progress 

Postoperative observations were made every 6 months to 1 year using either direct or rear oblique view endoscopy. Biopsies were performed when necessary. Patients diagnosed with adenoma upon repeat biopsy were considered as recurrence. 

During the study period, no clear guidelines regarding recurrence of ampullary tumors have been detailed; therefore, the attending physicians discussed and determined the choice of treatment.

### 2.3. Statistical Analysis

Data were analyzed using the statistical EZR software (version 1.54; ‘EZR’ (Easy R), Saitama, Japan) [5]. Student’s *t*-test or Mann-Whitney U test was used to compare categorical and continuous variables within groups. The log rank-test was used to evaluate the cumulative recurrence free rate between EP and TDA group.

## 3. Results

Patients’ background characteristics and therapeutic outcomes are shown in Table 1 and Table 2, respectively. No significant difference was observed regarding the age and sex between the EP and TDA groups. Tumor size was significantly larger in the TDA group compared to the EP group. No significant difference was observed regarding the preoperative diagnoses between the two groups; however, a significantly higher percentage of final diagnoses of adenocarcinoma was observed in the TDA group. Additionally, two patients from the EP group were finally diagnosed as having “normal epithelium.” Preoperative biopsies of these patients showed adenoma measuring 8 mm in one patient and adenocarcinoma (Tis) measuring 6 mm in the other. Both patients showed no recurrence during the follow-up period. 

Investigation of the resected samples showed that all samples from the TDA group were en bloc resections, and two patients from the EP group had their samples split into two. No significant difference was observed regarding the positive results of the lateral and vertical margins between the two groups. Figure 2 shows the distribution of the tumor sizes as related to lateral and vertical margins for both EP and TDA patients. No significant difference was observed regarding negative and positive/unevaluable margins in the two groups. 

Adverse events that occurred in the EP group were bleeding (three patients), mild pancreatitis (three patients), and bile duct stenosis (one patient; Table 2). In the TDA group, adverse events included mild pancreatitis (one patient), perforation (one patient), and intra-abdominal abscess (one patient). No significant difference was noted regarding the number of adverse events between the two groups. However, long-term hospitalization (over 30days) was observed in 2 cases in the TDA group. One patient was hospitalized for 58 days due to perforation and intra-abdominal abscess, and another patient was hospitalized for 38 days due to acute pancreatitis. This significantly increased the length of hospital stay in the TDA group.

The mean follow-up time was 36.5 months (EP), and 40.3 months (TDA). No significant difference was observed regarding the recurrence rate and interval until recurrence between the two groups (Table 2, Figure 3A). Three and two patients from the EP and TDA group, respectively, developed recurrence. The characteristics of the patients who developed recurrence are shown in Table 3. Two patients and one patient from the EP and TDA group, respectively, demonstrated either positive or unevaluable resection margins. However, one patient each from the EP and TDA groups developed recurrence despite negative margins following en bloc resection. Recurrences occurred in 7.4% (2/27) of patients with negative margins. No significant difference was observed regarding the relationship of recurrence with tumor size between the two groups (Figure 3B). The attending physicians discussed and determined the choice of treatments for recurrence of ampullary tumors. Argon plasma coagulation (APC), hot biopsy, radiofrequency ablation (RFA), and/or EP were applied (Table 3).

## 4. Discussion

In this study, we retrospectively compared the clinicopathological features and postoperative outcomes between EP and TDA groups. There were no significant differences in the therapeutic outcomes between the two groups except the shorter hospital stay in EP group.

The adenoma-carcinoma sequence is believed to be related to the malignant transformation of ampullary tumors, similarly observed in colorectal cancer [6,7,8]. A previous study found that the preoperative diagnostic accuracy is not high, particularly in its diagnosis of adenoma [9]. In the present study, the diagnostic accuracy rate was 83.3% (35/42). Therefore, EP and TDA implies a complete excision biopsy. Regarding cancer, lymph node metastasis does not occur in cases of Tis but occurs in pT1 in addition to micro-lymphatic invasion [10]. Lymph node metastasis is not rare in patients with T1b with sphincter of Oddi invasion, compared to T1a which are limited to ampullary mucosa. Trikudanathan et al. [11]. reported that the sensitivity (95%CI)/specificity (95%/CI) of EUS was 77% (0.69–0.83)/78% (0.72–0.84) for T1, 72% (0.65–0.80)/76% (0.71–0.83) for T2, 79% (0.71–0.85)/76% (0.71–0.83) for T3, and 84% (0.73–0.92)/74% (0.63–0.83) for T4, indicating poor diagnostic accuracy [12]. In EP and TDA indications, opinions regarding their sole indication for adenoma or the inclusion of Tis or T1a remain controversial. Difficulties in the preoperative diagnosis are expected. Previous studies have shown that tumors measuring until 50 mm are managed using EP [13,14]. However, perceptions regarding the correlation between tumor size and cancer remain controversial [15]. In the present study, no significance was observed regarding the relationship of tumor size and the final diagnosis of adenoma and adenocarcinoma (Figure 4). The TDA group had significantly larger tumors. This may be attributed to the concern regarding the difficult performance of EP when the tumor is laterally and widely spread or the endoscopic range of motion is restricted in the duodenum; in these cases, TDA was performed. Additionally, reports regarding the use of EP with hybrid-ESD in patients demonstrating superficial layer progress have recently been published [16]; we look forward to future research in this field.

Regarding the N and M factors, PD is indicated in patients with N1, and chemotherapy is indicated in patients with M1; MD-CT is the main diagnostic examination in both cases. Fong et al. found that among 41 patients with ampullary adenocarcinoma, MD-CT indicated lymphadenopathy in 10 patients, of whom, 5 were diagnosed lymph node metastases at pathology (50%). Furthermore, they found lymph node metastasis was found in 61.3% of the patients without lymphadenopathy on imaging [17]. Thus, even when the preoperative diagnosis is N0M0, in cases of T1a or deeper, the patient’s course needs to be carefully observed and PD or other additional therapies need to be considered. 

Heise et al. reported that the rate of complication was clearly higher in PD group than in EP and TDA groups [4]. Similarly, our investigation of treatment invasiveness indicated that there was no significant difference between EP and TDA regarding adverse events. The length of hospital stay was shorter, and the degree of invasiveness was lower in the EP group, which were consistent with those of a previous study [18].

A previous study observed that 33% of patients developed recurrence which was related to final diagnosis, intraluminal tumor presence, FAP complication, and experience of endoscopist [13]. Systematic review indicated that the recurrence rate was 13.0% in EP and 9.4% in TDA [4]. In the present study, we found that 9.1% (3/33) of the patients from the EP group and 22.2% (2/9) of those from the TDA group developed recurrence. Intraoperative frozen section was evaluated in only 2cases in the TDA group. This may be the reason for the relatively high recurrence rate in the TDA group, but there was no significance between the groups. No significant difference was noted in tumor size and recurrence (Figure 3B). All cases of recurrence were adenoma. Additionally, patients with negative margins in the resected samples suffered recurrence; particularly, in one patient, recurrence developed after 3 years. We believe that postoperative monitoring is essential even in patients with negative margins. Furthermore, careful monitoring and management are required since recurrence occurred after >4 years in one patient. 

The limitations of this study include the single-center location, the relatively small number of patients in the TDA group, the non-standardization of pathologic sample processing, and insufficient evaluation of the bile and pancreatic ducts in resected specimens.

Regarding the issue of the treatment indicated for ampullary tumors, EP can be the first-line treatment for adenomatous lesions, because it is associated with less degree of invasiveness and does not have a poor clinical outcome. However, when performing EP with technical difficulties, such as in cases of large tumor size, we consider the use of TDA. This does not deviate from the ESGE guideline [3]. We believe that EP with hybrid-ESD should be considered in patients who are unable to tolerate surgery and general anesthesia. 

## 5. Conclusions

In cases of ampullary tumors, it is prudent to consider the possibility of adenocarcinoma as the final diagnosis even if preoperative biopsy indicates adenoma, regardless of tumor size. No significant difference was observed in the therapeutic outcomes of EP and TDA, except hospital stay; therefore, minimally invasive EP is initially considered. TDA is considered as an option based on tumor size and other factors. Recurrence may occur even in patients with negative margins; therefore, careful monitoring during the postoperative course is necessary. 

## Figures and Tables

**Figure 1 jcm-10-04463-f001:**
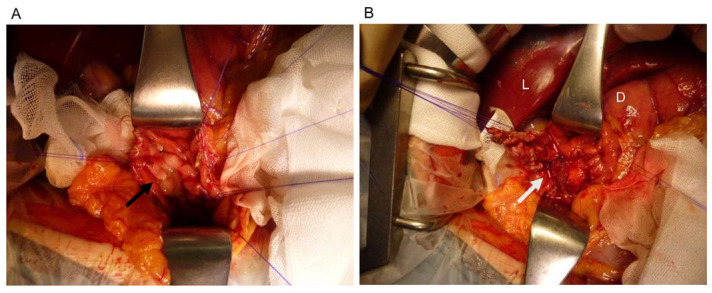
Intraoperative image of TDA. (**A**) After duodenotomy and before TDA, the ampullary tumor was visualized (black arrow). (**B**) The ampullary tumor was resected (white arrow). L: Liver, D: Duodenum.

**Figure 2 jcm-10-04463-f002:**
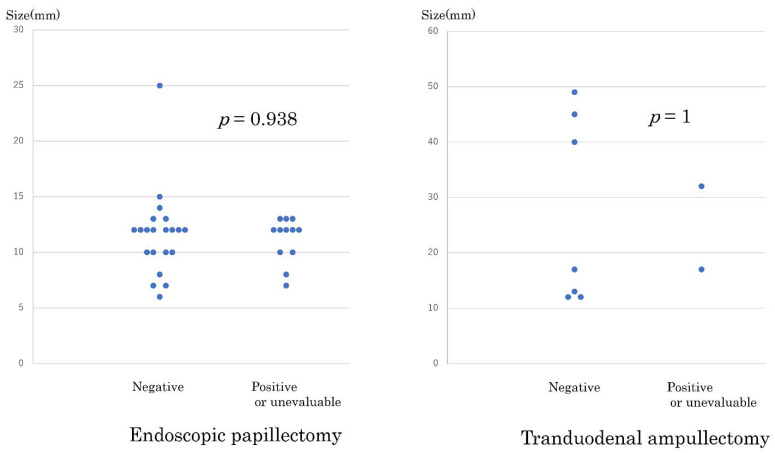
Distribution of tumor sizes related to negative and positive/unevaluable margins. EP (endoscopic papillectomy) (**left**), TDA (transduodenal ampullectomy) (**right**).

**Figure 3 jcm-10-04463-f003:**
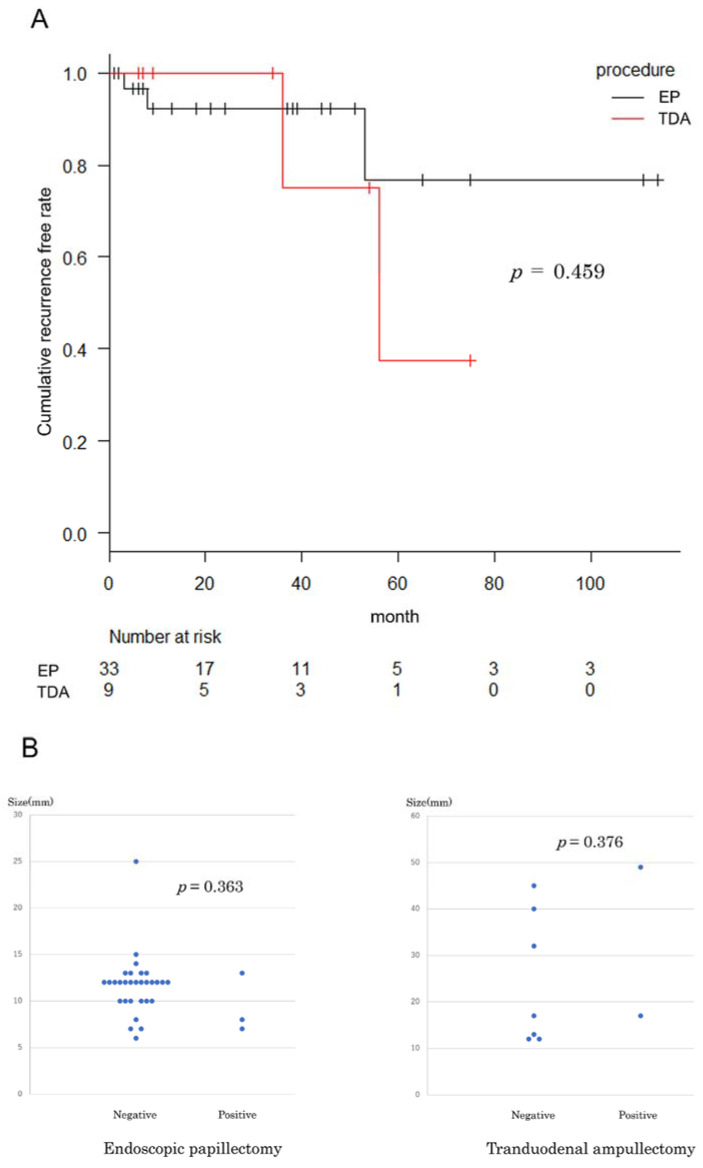
Comparison of recurrence between EP and TDA group. (**A**) Cumulative recurrence free rate. (**B**) Distribution of tumor sizes related to recurrence (positive vs. negative): EP (**left**), TDA (**right**).

**Figure 4 jcm-10-04463-f004:**
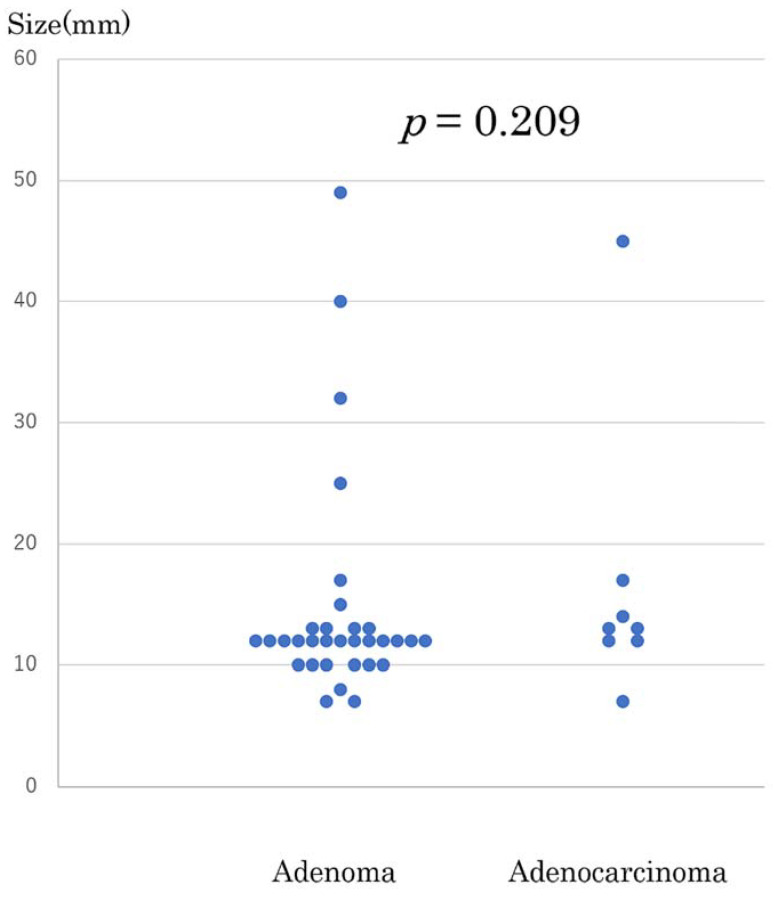
Distribution of tumor sizes according to final diagnoses of adenoma and adenocarcinoma. Adenoma (**left**), Adenocarcinoma (**right**).

**Table 1 jcm-10-04463-t001:** Patients’ background characteristics.

	Total (42)	EP (33)	TDA (9)	*p*-Value
Gender				0.0805
Male	28	24	4	
Female	14	9	5	
Age (years median range)	67.7 (31–83)	67.9 (44–81)	66.8 (31–83)	0.975
Tumor size (mm median range)	14.6 (6–49)	11.5 (6–25)	26.3 (12–49)	0.0196
Preoperative diagnosis				0.347
Adenoma	39	32	7	
Adenocarcinoma	3	1	2	
Extensive intraepithelial progress in the common bile duct				0.195
Nagative	37	30	7	
Positive	1	0	1	
Unevaluable	4	3	1	
Extensive intraepithelial progress in the main pancreatic duct				0.374
Nagative	38	30	8	
Positive	0	0	0	
Unevaluable	4	3	1	

**Table 2 jcm-10-04463-t002:** Post-treatment outcomes.

	Total (42)	EP (33)	TDA (9)	*p*-Value
Postoperative diagnosis				**0.0353**
Adenoma	32	26	6	
Adenocarcinoma	8	5	3	
Normal epithelium	2	2	0	
En block resection				NA
Yes	40	31	9	
No	2	2	0	
Lateral margin				0.195
Negative	33	26	7	
Positive	3	2	1	
Unevaluable	6	5	1	
Vertical margin				0.195
Negative	30	23	7	
Positive	3	3	0	
Unevaluable	9	7	2	
Adverse event	6	7	3	0.594
Bleeding	3	3	0	
Mild pancreatitis	2	3	1	
Bile duct stenosis	1	1	0	
Perforation	1	0	1	
Intra-abdominal abscess	1	0	1	
Mortarity	0	0	0	
Duration of hospitalization (day, mean, range)	15.7 (8–52)	13.6 (8–28)	23.4 (13–52)	**0.0471**
Follow-up period (month, mean, range)	37.4 (1–114)	36.5 (1–114)	40.3 (6–96)	0.587
Recurrence	5	3	2	0.169
Time to recurrence (month, mean, range)	31.2 (3–56)	21.3 (3–53)	46 (36–56)	0.169

Bold: significant differences.

**Table 3 jcm-10-04463-t003:** Characteristics of patients who developed recurrence.

Tumor Size (mm)	Preoperative Diagnosis	Extensive Intraepithelial Progress in the Common Bile Duct	Extensive Intraepithelial Progress in the Main Pancreatic Duct	En Block Resection	Postoperative Diagnosis	Lateral Margin	Vertical Margin	Time to Recurrence (Month)	Treatment for Recurrence
8	Adenoma	Negative	Negative	Yes	Adenoma	Unevaluable	Unevaluable	53	APC
17	Adenoma	Positive	Negative	Yes	Adenoma	Unevaluable	Negative	56	APC, RFA
13	Adenoma	Negative	Negative	No	Adenoma	Negative	Unevaluable	8	EP, Hot biopsy
7	Adenoma	Negative	Negative	Yes	Adenoma	Negative	Negative	3	EP
49	Adenoma	Negative	Negative	Yes	Adenoma	Negative	Negative	36	Hot biopsy

## Data Availability

Data available in a publicly accessible repository.

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
