# Peer review of "Investigation of the Indications for Endoscopic Papillectomy and Transduodenal Ampullectomy for Ampullary Tumors"

_jcm, 2021, doi:10.3390/jcm10194463_

Round 1
Reviewer 1 Report
Masanari Sekine et al in the prensent manuscript evaluated the effectiveness of EP and TDA for ampullary tumors. They evaluated retrospectively 42 patients who underwent either endoscopic EP (33) or TDA (9) for ampullary tumors during 9 years. They conluded for no significant relationship between tumor size and recurrence. The paper is of interest however deserve some major changes:
1- The major limitation of the manuscript in the present form is the low number of pts with TDA included. Such bisas might influence the results in term of morbidity analysis
2- Was the intraoperative frozen sections evaluated expecially for TDA? When you perform a TDA is crucial to have a frozen section analysis. How do you explain the 22% of recurrence in the TDA group?
3- 23 day median range of post operative hospital stay it look a bit too long. How do you explain? Was a fast track program used?
Minor comments
1- I wonder that you can performe a radical TDA with a 2 cm incision
2- Please better describe the TDA procedure. Was the wirsung duct isolated and reconstructed separatly or with the CBD?
3- Plaese evaluated the cumulative morbidity between the two groups in table 2
4- Was the case decision making made in a multidisciplinary setting?
5- What have you done in the pts who developed recurrence? Please add in tacble 3
6- To have a sigle or doble intraoperative picture of TDA would be of interest for the readers
7- Figure 3 can be omitted
Author Response
Major
- The major limitation of the manuscript in the present form is the low number of pts with TDA included. Such bias might influence the results in term of morbidity analysis
We totally agree with your opinion. As the technique of endoscopic procedures progressed, the number of EP increased, but the number of TDA did not increase. Due to the strict revision time limit (5 days), it is not possible to collect any more cases or to collaborate with other centers. We clearly described this limitation in the Discussion section (P.14 l.15-16).
- Was the intraoperative frozen sections evaluated especially for TDA? When you perform a TDA is crucial to have a frozen section analysis. How do you explain the 22% of recurrence in the TDA group?
Intraoperative frozen sections were evaluated in only 2 cases in the TDA group. The duodenal mucosa was incised at least 5mm from tumor and ampulla tumor was resected with careful identification of the sphincter of Oddi. But in one recurrent case in the TDA group, the lateral margin was unevaluable. We should have evaluated the frozen section in all cases. This may be the reason for the relatively high recurrence rate in the TDA group, as you pointed out. We added the description in the Discussion section (P.14 l.5-9).
- 23 day median range of post operative hospital stay it look a bit too long. How do you explain? Was a fast track program used?
Two patients from the TDA group were hospitalized for a long-term. One patient was hospitalized for 58 days due to perforation and intra-abdominal abscess, and another patient was hospitalized for 38 days due to acute pancreatitis. We did not use a fast track program. We added the description in the Result section (P.9 l.17-18, P10 l .1-8).
Minor
1.I wonder that you can perform a radical TDA with a 2 cm incision.
A 2-4 cm longitudinal duodenotomy was performed. When the tumor size is less than 15 mm, a 2 cm incision is enough to perform a radical TDA.
2.Please better describe the TDA procedure. Was the wirsung duct isolated and reconstructed separatly or with the CBD?
We added the detail of the TDA procedure for better understanding (P.7 l.12-15). The Wirsung duct was not isolated and reconstructed. To repair the cavity of the lost mucosa, the mucosa and the sphincter of Oddi were radially sutured to prevent obstruction of the Wirsung duct and the common bile duct.
3.Plaese evaluated the cumulative morbidity between the two groups in table 2
We added Figure 3A to show the cumulative recurrence free rate.
4.Was the case decision making made in a multidisciplinary setting?
During the study period, no clear guidelines regarding ampullary tumors have been published; therefore, the attending physicians discussed and determined the choice of EP or TDA. We described in P.5 l.13-16.
5.What have you done in the pts who developed recurrence? Please add in table 3
During the study period, no clear guidelines regarding treatment for recurrence of ampullary tumors have been detailed; therefore, the attending physicians discussed and determined the choice of treatments. We added the description in P.8 l.5-7 and a column to show the treatment for recurrence in Table 3.
6.To have a sigle or doble intraoperative picture of TDA would be of interest for the readers
We added two intraoperative pictures before and after TDA as Figure 1.
7.Figure 3 can be omitted
We omitted Figure 3.
Reviewer 2 Report
This manuscript compares the indications and outcomes of endoscopic papillectomy (EP) and transduodenal ampullectomy (TDA) for ampullary tumors.
My greatest concern lies within the role of EP and TDA. Are they interchangeable? As the authors state in their discussion, EP is the first choice and TDA is a secondary choice to EP when EP is not feasible. So is it a reasonable comparison? TDA is a very invasive procedure requiring deuodenotomy. Therefore, when a benign ampullary tumor is suspected, it is not a decision making between EP and TDA. TDA is reserved in case EP is not possible. In this regard, I do not see the point in comparing EP and TDA.
Second, the number of TDA is too small for a statistical comparison. The results can be easily over-turned with larger numbers. I would recommend collecting more cases, or conducting a collaborative study with other centers.
Lastly, for improvement, I would recommend to conduct a study comparing short and long-term outcome between EP (+/-TDA) and standard operation groups instead of comparing EP and TDA which are in rather complementary relationship. That could point to safety and feasibility of local resection and suggest which tumors are safe to recommend local excision using either EP (+/- TDA).
Author Response
- My greatest concern lies within the role of EP and TDA. Are they interchangeable? As the authors state in their discussion, EP is the first choice and TDA is a secondary choice to EP when EP is not feasible. So is it a reasonable comparison? TDA is a very invasive procedure requiring deuodenotomy. Therefore, when a benign ampullary tumor is suspected, it is not a decision making between EP and TDA. TDA is reserved in case EP is not possible. In this regard, I do not see the point in comparing EP and TDA.
No clear guidelines were published for ampullary tumors during the study period. Surgeons tended to choose TDA and endoscopists tended to choose EP. TDA is a radical and invasive procedure and EP is less invasive but may have some disadvantages, such as low R0 rate. Therefore, we retrospectively compared these two treatments. We described the issue clearly in the Introduction section. With the results in this study, we believe that EP can be the first choice and TDA is a secondary choice when EP is not feasible. Recently, there are reports concerning endoscopic papillectomy with hybrid endoscopic submucosal dissection. ESD is a technique standing between EP and TDA and the boundary of indication between EP and TDA might be disappearing.
- Second, the number of TDA is too small for a statistical comparison. The results can be easily over-turned with larger numbers. I would recommend collecting more cases, or conducting a collaborative study with other centers.
We totally agree with your opinion. As the technique of endoscopic procedures advanced, the number of EP increased, but the number of TDA did not increase. Due to the strict revision time limit (5 days), it is not possible to collect any more cases or to collaborate with other centers. We clearly described this limitation in the Discussion section (P.14 l.15-16).
3.Lastly, for improvement, I would recommend to conduct a study comparing short and long-term outcome between EP (+/-TDA) and standard operation groups instead of comparing EP and TDA which are in rather complementary relationship. That could point to safety and feasibility of local resection and suggest which tumors are safe to recommend local excision using either EP (+/- TDA).
For the clinically diagnosed adenocarcinoma, we basically performed a standard operation (PD). For the clinically diagnosed adenomatous lesions, we performed EP or TDA. There were no PD cases for ampullary adenomatous lesions, we could not compare EP (+/-TDA) and standard operation.
Reviewer 3 Report
The manuscript aims to evaluate the effectiveness of endoscopic papillectomy (EP) and transduodenal 16 ampullectomy (TDA) for for ampullary tumors
A single center experience with 42 patients is desccribed. Given the novelty of endoscopic treatment of ampullomas and the small series published the topic is of interest. The manuscript describes good techniques and data is informative since histology and margins as well as follow up iasa adequate.
In introduction some recent publications regarding this topic could be referred to and some additional information could be given. References could include
Geoffroy Vanbiervliet et al
Endoscopic management of ampullary tumors: European Society of Gastrointestinal Endoscopy (ESGE) Guideline Endoscopy 2021; 53 and
Systematic Review with Meta-Analysis: Endoscopic and Surgical Resection for Ampullary Lesions. J Clin Med. 2020 Nov 10;9(11):3622. doi: 10.3390/jcm9113622.PMID: 33182806.
Methods should be better described. Especially the questions: How long was follow up? (including median and range for the total cohort and different groups). Are there any differences in length of follow up between EP and TDA?
Describe how information was retrieved (medical records?) Any patients lost to follow up?
Is mortality in hospital mortality or 30 days mortality?
Discussion: withdraw figure 3 and refer to ESGE guidelines in introduction. Instead discuss your results (complications, recurrence etc) in relation to published studies although there are not son many, there are still some.
Author Response
In introduction some recent publications regarding this topic could be referred to and some additional information could be given.
Thank you very much for pointing out valuable literature. We cited the references and rewrote the Introduction section.
2.Methods should be better described. Especially the questions: How long was follow up? (including median and range for the total cohort and different groups). Are there any differences in length of follow up between EP and TDA?
We added the detail of the TDA procedure for better understanding (P.7 l.12-15). We added the description of follow-up period in the Result section (P.10 l.9) and added a column to show the follow-up period (mean, range) to Table2. There were no differences in length of follow up between EP and TDA groups.
- Describe how information was retrieved (medical records?) Any patients lost to follow up?
The information of patients was retrieved from medical records. We added the description in the Result section (P.5 l.4). We added the cumulative recurrence free rate in Figure 3A. Seventeen patients were referred to other hospitals, 6 patients dropped out of our hospital, and one patient died of other illness.
- Is mortality in hospital mortality or 30 days mortality?
Mortality is 30 days mortality. We added the description in the Result section (P.5 l.4-5).
5.Discussion: withdraw figure 3 and refer to ESGE guidelines in introduction. Instead discuss your results (complications, recurrence etc) in relation to published studies although there are not son many, there are still some.
We omitted Figure 3. We rewrote the Introduction section and referred to ESGE guideline. In the Discussion section, we cited the literature to discuss our clinical outcomes (complication, recurrence).
Round 2
Reviewer 1 Report
the Authors in the revided version
provided Point to Point clear answers
Reviewer 2 Report
The authors have adequately answered to all reviewers' comments and made necessary modifications to improve their manuscript.